# The Dark Side of Iron: The Relationship between Iron, Inflammation and Gut Microbiota in Selected Diseases Associated with Iron Deficiency Anaemia—A Narrative Review

**DOI:** 10.3390/nu14173478

**Published:** 2022-08-24

**Authors:** Ida J. Malesza, Joanna Bartkowiak-Wieczorek, Jakub Winkler-Galicki, Aleksandra Nowicka, Dominika Dzięciołowska, Marta Błaszczyk, Paulina Gajniak, Karolina Słowińska, Leszek Niepolski, Jarosław Walkowiak, Edyta Mądry

**Affiliations:** 1Department of Pediatric Gastroenterology and Metabolic Diseases, Poznan University of Medical Sciences, 61-701 Poznan, Poland; 2Department of Physiology, Poznan University of Medical Sciences, 61-701 Poznan, Poland

**Keywords:** inflammatory bowel disease, colorectal cancer, obesity, oxidative stress, dysbiosis, anaemia

## Abstract

Iron is an indispensable nutrient for life. A lack of it leads to iron deficiency anaemia (IDA), which currently affects about 1.2 billion people worldwide. The primary means of IDA treatment is oral or parenteral iron supplementation. This can be burdened with numerous side effects such as oxidative stress, systemic and local-intestinal inflammation, dysbiosis, carcinogenic processes and gastrointestinal adverse events. Therefore, this review aimed to provide insight into the physiological mechanisms of iron management and investigate the state of knowledge of the relationship between iron supplementation, inflammatory status and changes in gut microbiota milieu in diseases typically complicated with IDA and considered as having an inflammatory background such as in inflammatory bowel disease, colorectal cancer or obesity. Understanding the precise mechanisms critical to iron metabolism and the awareness of serious adverse effects associated with iron supplementation may lead to the provision of better IDA treatment. Well-planned research, specific to each patient category and disease, is needed to find measures and methods to optimise iron treatment and reduce adverse effects.

## 1. Introduction

Iron deficiency (ID) and iron deficiency anaemia (IDA) are important causes of diseases and disabilities worldwide. In 2016 there were over 1.2 billion cases of IDA, of which 41.7% of children (younger than 5 years), 40.1% of pregnant women, and 32.5% of non-pregnant women were anaemic worldwide; thus, IDA is one of the world’s primary public health issues (Figure 1), [1,2,3]. For the World Health Organization (WHO), controlling anaemia is a global health priority with the purpose of a 50% reduction in anaemia prevalence in women by 2025 [4]. IDA mainly affects premenopausal and pregnant women, growing children and the elderly, however, it is also increasingly recognised as a complication of multiple diseases associated with lifestyle and dietary patterns, such as obesity, inflammatory bowel disease, colorectal cancer or chronic kidney disease [5,6].

There are two goals of the IDA treatment, to replenish iron stores and normalise haemoglobin concentration. Individual assessment of IDA causes and amount of dietary iron intake is recommended. The holistic nutritional strategy to treat and avoid iron deficiency in the future includes the two paths regarding iron intake and absorption. First is to increase the iron intake by promoting the consumption of iron-rich food such as meat, poultry, fish, and seafood. To optimise iron absorption it is recommended to increase vitamin C and fermented product intake and avoid the consumption of iron absorption inhibitors (phytates in cereals and nuts, tannins from coffee or tea and calcium-rich products) together with iron-rich food [7]. Although a described holistic approach to clinical management of ID is available, oral or parenteral iron products with varying doses and formulations are the primary means of supplementation [8]. However, there is a growing body of evidence for side effects of iron supplementation concerning the exacerbation of inflammation, alterations in gut microbiota, and gastrointestinal adverse events [9]. Therefore, this review aimed to investigate the knowledge regarding the relationship between iron supplementation, inflammatory status and changes in the gut microbiota milieu in diseases typically complicated with IDA and considered as having an inflammatory background, including inflammatory bowel disease, colorectal cancer, and obesity.

## 2. Search Strategy and Selection Criteria

PubMed/Medline and Cochrane databases were searched for articles about iron deficiency anaemia, iron and its relationship to inflammatory bowel disease, colorectal cancer and obesity. The following English terms and their combinations were used: iron, iron metabolism, iron deficiency, iron deficiency anaemia, microbiota, inflammatory bowel disease, colorectal cancer, colon cancer, and obesity. The titles and abstracts were screened for the 282 papers, and 97 were chosen for thorough study. In addition, a review of their reference lists identified further relevant articles, so in total, 141 articles were selected for review.

### 2.1. Iron in the Human Body

Iron is a micronutrient fundamental for many biochemical reactions. It acts primarily as a component of enzymes or other proteins involved in oxygen transport, particularly in haemoglobin and myoglobin. Moreover, it is involved in energy metabolism in mitochondria, cell proliferation and growth, and DNA processing [10,11]. In contrast to other nutrients, iron has no active excretory mechanisms; therefore, its resource control must be provided in the small intestine [12]. Iron is lost mainly with shedded enterocytes, epidermic cells, and menstrual blood in women, and less with urine or faeces [10]. Most body iron stores are regained from senescent and damaged erythrocytes as an endogenous source [13]. Due to the modest iron loss, absorbing 1 to 2 milligrams daily is enough to compensate for iron decline and maintain a concentration of 3–5 g [10,14].

Dietary iron heme is provided in the form of myoglobin and haemoglobin in animal food sources or as non-heme iron in plant and animal food sources [13]. Due to the difference in the biochemical structure, heme and non-heme iron vary in bioavailability. Non-heme iron tends to sequestrate and bind to other food components and this causes lower absorption in comparison to heme iron [13,15,16]. Although heme iron does not make up the majority of iron diet intake in carnivores, this form is most easily absorbed [13]. Iron deficiency and excess lead to disordered homeostasis, with excess iron increasing the pool of labile iron in plasma. Unbound iron causes an overload in tissues and further damage by generating reactive oxygen species (ROS) via the Fenton or Haber-Weiss reactions [17]. Subsequently, lipid and amino acid peroxidation and DNA damage occur leading to cell death [10,18].

### 2.2. Iron Absorption and Metabolism

Iron absorption occurs mainly in the duodenum and proximal part of the jejunum, which is mostly adjusted to this role due to the high expression of proteins involved with non-heme iron absorption [15,16,19]. Regarding heme iron, there are two prevailing hypotheses of its absorption: the receptor-mediated endocytosis of heme and direct transport into the intestinal enterocyte by heme transporters [20]. Non-heme iron uses divalent metal-ion transporter-1 (DMT1) to enter the apical membrane of the enterocyte. DMT1 function is conjugated with the electrochemical gradient of protons that generate the driving force for iron transfer from the extracellular space to the cytoplasm of the cells of the intestinal lining [21,22]. The importance of the DMT1 symporter was confirmed in animal models with an intestinal DMT1 knockout that caused a severe iron absorption disorder [23]. Furthermore, a rare DMT1 mutation was also detected in humans with microcytic hypochromic anaemia [24]. The DMT1 function must be assisted by an enzyme called duodenal cytochrome B (DcytB) located in the brush border [15].

Iron exists in an oxidised, insoluble ferric (Fe^3+^) form at physiological pH and to be absorbed, it must be reduced into a soluble ferrous (Fe^2+^) state or be bound by a protein, such as heme [25]. Hence, non-heme iron is transported via DMT1 in the form of Fe^2+^ ions. As DcytB executes the non-heme iron reduction from Fe^3+^ to Fe^2+^, it forms a substrate for DMT [14,25]. Higher DcytB expression in the enterocyte lining leads to increased iron uptake, which indicates DcytB’s major role in iron absorption [26].

If iron is not currently required for any ongoing processes, it is temporarily housed in ferritin. Ferritin is a cytosolic protein capable of sheltering up to 4500 Fe3^+^ atoms in one molecule [15,27]. Increased iron demand activates ferroportin, the transmembrane protein located in the basolateral side of duodenal enterocytes. It facilitates the transfer of enterocyte iron stores into the bloodstream, where iron is bound by transferrin. This step needs iron to be oxidised back to Fe^3+^, which is catalysed by the enzyme hephaestin. The iron–transferrin complex is then recognised by transferrin receptors leading to iron endocytosis in the body cells and subsequent iron use in cellular processes [15,28].

Iron metabolism is precisely regulated both on systemic and cellular levels. The first one involves mainly a liver-derived peptide, hepcidin, the expression of which depends on the body’s iron demand [15]. When the organism is saturated with iron, hepcidin expression increases and it interacts with ferroportin to cause ferroportin internalisation and subsequent degradation, which decreases iron transfer into the bloodstream. In contrast, the amount of hepcidin decreases during ID and facilitates the alignment of iron stores by unimpeded ferroportin function [29].

The cellular iron regulation is based mainly on two regulatory proteins—iron regulatory protein 1 (IRP1) and iron regulatory protein 2 (IRP2). They are responsible for post-transcriptional regulation of the iron homeostatic gene expression. The IRP action mechanism involves binding to iron-responsive elements (IREs) in the untranslated target mRNA regions. So, IRPs control the translation and stability of mRNA for proteins related to iron, oxygen and energy metabolism. Interestingly IRP2 can also act as cytosolic aconitase. IRP1 controls the expression of hypoxia-inducible factor 2α (HIF2α). Thus, it is considered a key regulator of erythropoiesis. IRP2 regulates the expression of transferrin receptor 1 (TfR1) and 5-aminolevulinic acid synthase 2. It is responsible for iron uptake and heme biosynthesis in erythroid progenitor cells (for further information, see [15,30]). The main aspects of iron metabolism in the human body are presented in Figure 2.

### 2.3. Iron Toxicity

Iron is both essential and potentially harmful. Therefore, its homeostasis requires precise regulatory mechanisms to provide cells with the necessary element supply and concurrently prevent the unfavourable effects of excess iron [17,31,32]. Transferrin and ferritin bind and store iron, respectively, to protect against the adverse effects of labile iron [31]. Iron toxicity occurs because the iron-binding protein capacity is restricted. Its toxicity stems from two distinct qualities of the metal, the ability to generate free radicals and serving as an essential growth factor for nearly all pathogenic bacteria, fungi, and protozoa, as well as for all neoplastic cells [33]. It is also worth mentioning that excess dietary iron may affect the ingestion of other divalent metals. For example, in murine and rat models, the high nutritional iron intake contributed to copper deficiency [34,35,36,37]. However, the clinical significance of this finding in humans requires further investigation.

### 2.4. Iron and Oxidative Stress

A parameter that reflects excessive iron accumulation is a pule pool of labile iron, which physiologically accounts for about 5% of total cellular iron. Iron overload increases the non-transferrin-bound iron. Unbound iron is particularly hazardous because it readily accepts and donates electrons while switching between the two forms, Fe^2+^ and Fe^3+^, leading to free radical generation [38]. Hence, “free iron” is highly reactive and can participate in Fenton’s or Haber-Weiss’s reactions, generating ROS, such as highly dangerous hydroxyl radicals [39]. ROS initiate lipid and amino acid peroxidation, enzyme denaturation, polysaccharide depolymerisation and DNA damage [39,40]. An insufficiency of antioxidative mechanisms due to iron excess and oxidative stress triggers cell damage [39,41,42,43]; therefore, iron must be sequestered by ferritin or transferrin to prevent ROS production [28,39].

Since the small intestine is the leading site for iron absorption and regulation, ROS-generating responses primarily affect this part of the gastrointestinal tract. Overproduction of free radicals causes cellular changes and increases intestinal barrier permeability. These changes are exacerbated by the onset of inflammation as expressed by hepcidin and/or calprotectin levels, which increase with oral iron supplementation [44,45,46]. ROS generate stress in the mitochondria and endoplasmic reticulum, causing swelling. This is the first step in the initiation of the cell death pathway [47]. Moreover, this process is facilitated by the increased peroxidation of lipids leading to the destruction of cell membranes [48]. Enterocyte apoptosis finally violates gut barrier function and contributes to the increased permeation of substances such as bacterial toxins into the bloodstream [47].

These mechanisms were demonstrated in Chinese Yellow broilers treated with a high dose of dietary iron, leading to increased malondialdehyde (MDA), the main product of lipid peroxidation. Alterations within intestinal villi were also reported [49]. The research on rats also demonstrated adverse intestinal effects of the iron supply in the context of oxidative stress [17]. The connection between iron-dependent oxidative stress and the aggravation of inflammation from the viewpoint of disorders and diseases is described in detail below.

### 2.5. Iron and Microbiota

Although commonly used, iron supplementation may have adverse effects on the gastrointestinal tract, mainly on the intestinal microbiota [50]. This article focuses mainly on the mutual interaction between the oral iron supply and gut microbiota. The excess unabsorbed iron passes through the colon and is involved in Fenton and Haber-Weiss reactions, with adverse effects on the intestinal structure. Iron supplementation also affects the microbiota [51], with the composition of intestinal flora affecting iron absorption [52]; therefore, iron and microbiota are in a complex and bilateral relationship [53].

The amount of available “free iron” in the intestines is significantly lower than the optimal level required for the proper functioning and replication of bacterial cells [52,54,55,56]. Some pathogenic bacteria like *Salmonella* or the pathogenic strain of *Escherichia coli* are equipped with siderophores, extracellular ferric chelators that enable bacteria to capture trivalent iron (Fe^+3^) from ferritin and transferrin [56,57]. Siderophores are considered virulence factors, as they help bacteria to survive in an iron-deficient environment [58]. Interestingly, some commensal bacteria such as the genera *Lactobacillus* and *Bifidobacterium* do not require a high portion of iron to grow and expand [59].

The intestinal microbiota controls systemic iron homeostasis in two ways, first, by inhibiting the intestinal iron absorption pathways via hypoxia-induced factor (HIF-2α). In a state of increased iron demand (ID, hypoxia, augmented erythropoiesis), HIF-2α increases the expression of DMT1, DcytB at the enterocyte apical brush border membrane, and ferroportin at the basal membrane [60,61,62]. Second, iron homeostasis is controlled by increasing cellular iron storage induced by higher ferritin expression [63]. Moreover, it was demonstrated that *Lactobacillus plantarum* increases iron absorption in the intestine of iron-deficient females [64], so could be used as a component of iron supplements to alleviate the negative side effects of oral iron administration [65]. Some studies have shown the reduction of the abundance of beneficial microbes simultaneously with an increased abundance of deleterious microbes after oral iron supplementation (Table 1), [66]. For example, African children with anaemia, fed with iron-fortified biscuits for six months, presented an unfavourable ratio of pathogenic to commensal bacteria and increased calprotectin concentration in the stools, which is indicative of intestinal inflammation. [67]. The other authors suggested that iron supplementation at the physiological level does not lead to mucositis unless other factors such as pathogen invasion or systemic inflammation are present [59]. Similar results concerning changes in microbiota were also obtained by studies conducted in children from Kenya, which is also suggestive of the negative impact of excess dietary iron on the intestinal microbiome [68,69].

However, some researchers found an entirely diverse outcome in rat experiments, with oral iron supplementation promoting commensal and dominant bacteria, and increasing the intestinal microbiota metabolic activity as evidenced by the increased concentration of beneficial short-chain fatty acids in the colon [70]. Furthermore, in a randomised controlled trial in South African school-aged children, no significant changes in the intestinal microbiome or inflammatory markers were observed due to oral iron supplementation [71].

In summary, studies concerning the relationship between microbiota and iron have shown inconsistent results, reporting negative and positive impacts, highlighting the need for further investigation on this topic. In particular, there is a lack of high-quality data examining the potentially harmful effect of untargeted iron supplementation, for example, in women of childbearing age [50,72].

## 3. Inflammatory Bowel Diseases (IBDs)

IBDs, including ulcerative colitis (UC) and Crohn’s disease (CD), are chronic, relapsing inflammatory conditions of the gastrointestinal (GI) tract. Interactions between the environmental factors and commensal intestinal microflora in genetically predisposed individuals are considered the leading cause of an inappropriate immune response and as a result, the development of inflammatory disease [73]. In recent years, the incidence of IBD has increased in highly industrialised western countries, mainly affecting people between 16 and 30 years old. It is associated with lifestyle changes, including a Western-style high-fat diet, cigarette smoking, distress, as well as taking oral contraceptives, hormone replacement therapy, and non-steroidal anti-inflammatory drugs [74].

UC primarily affects the colon and the rectum, whereas CD can involve any part of the GI tract, from the mouth to the anus, but most commonly involves the distal part of the small intestine and colon. IBDs may present with symptoms such as abdominal cramps and pain, persistent diarrhoea, fatigue, weight loss or bleeding.

### 3.1. Anaemia as a Complication of IBD

Anaemia is the most prevalent extraintestinal complication of IBD, with an estimated 70% of inpatients and 20% of outpatients with IBD developing anaemia, which is believed to affect one-third of the IBD patients at any one time [75]. Traditionally, anaemia was classified as IDA and anaemia of chronic disease (ACD). In IBD patients, both the aforementioned types of anaemia may occur because the local inflammation in the intestines contributes to the onset of systemic inflammation [76].

IDA is a consequence of chronic haemorrhages from the ulcerated mucosa, impaired dietary iron absorption, as well as self-imposed dietary restrictions relating to gastrointestinal symptoms. Moreover, appetite loss during an exacerbation of the disease and a range of other factors such as medicines used for IBD treatment (e.g., proton pump inhibitors, methotrexate, thiopurines and sulfasalazine) also negatively impact iron absorption and erythropoiesis [75]. In turn, ACD is connected with the systemic immune response that accompanies inflammatory diseases such as IBD [77]. Immune cells release pro-inflammatory cytokines, mainly interleukin-6 (IL-6), which upregulates the expression of liver hepcidin, the main regulator of iron homeostasis, thereby decreasing iron uptake from the enterocytes and a reduced ability to take advantage of sufficient iron for effective erythropoiesis [78]. This issue is described in more detail in the section about iron metabolism.

### 3.2. Iron Replacement Therapy in IBD

As described in the previous paragraph, the high incidence of anaemia in patients with IBD requires therapeutic intervention, either intravenous or oral iron administration. The European Crohn’s and Colitis Organization [79] guidelines recommend intravenous iron supply as a mainstay treatment for IBD patients and front-line therapy for haemoglobin (Hb) levels < 10 g/dL (e.g., iron sucrose, ferric gluconate, ferric carboxymaltose, iron isomaltoside). Another indication for intravenous iron is the active phase of the disease because inflammation impairs iron absorption in the intestines [80]. An intravenous iron supply is also recommended in patients with a poor iron tolerance and a previous unsuccessful attempt at oral iron treatment [81,82]. If oral iron supplementation must be used, it should be limited to IBD patients with mild anaemia (Hb ≥ 11.0 g/dL), with an inactive disease and no prior intolerance to oral iron [83].

### 3.3. Negative Consequences of Oral Iron

Oral iron administration can cause side effects due to a large amount of non-absorbed iron (about 90%) remaining in the intestines, including gastroduodenitis, nausea, bloating, vomiting, dyspepsia, constipation, diarrhoea, abdominal pain or darkening of the stools [47]. Thus, oral iron therapy is not optimal with as many as 50% of patients discontinuing the therapy [8]. The aforementioned gastrointestinal symptoms most likely result from a combination of a high concentration of free iron radicals induced by redox cycling in the gut lumen and at the mucosal surface which can promote inflammation and alterations in the gut microbiota composition [9,45].

Moreover, non-absorbed iron can be toxic and exacerbates disease activity in IBD. A study conducted in rats with dextran sulphate sodium (DSS)-induced colitis indicated that dietary iron administration aggravates colitis and is associated with oxidative stress, neutrophil infiltration and NF-kappaB pathway activation which increases the expression of pro-inflammatory cytokines such as interferon-γ (IFN-γ), tumour necrosis factor α (TNF-α), and inducible nitric oxide synthase (iNOS). These negative effects can be ameliorated by vitamin E [84]. Another animal study showed that a diet without iron sulphate combined with intravenous iron administration prevents the development of chronic ileitis in a mouse model of CD, suggesting that oral iron sulphate replacement therapy may trigger the inflammatory processes linked to the progression of CD-like ileitis [85]. There is an imbalance between pro-oxidative and antioxidant mechanisms in CD, with patients having increased levels of reactive oxygen intermediates (ROI) and DNA oxidation products, as well as markedly raised iron levels in combination with reduced copper and activity of zinc superoxide dismutase (Cu/Zn SOD) [86]. Oral iron due to its pro-oxidative capacity may promote a pro-inflammatory effect.

### 3.4. Impact of Iron on the Intestine

Iron has a significant impact on intestinal functioning, potentially via various mechanisms. Experimental evidence suggests that excessive iron in the lumen may be also harmful to the intestinal mucous membrane. Despite its key role in cellular processes, free iron in the colon can generate toxic free radicals that may directly impair the integrity of the intestinal epithelium via oxidative stress. The relationship between iron and redox stress is discussed in more detail in the section about iron and redox stress.

A key element of the physical intestinal barrier is a single layer of epithelial cells, mainly consisting of enterocytes, which in addition to absorbing nutrients, play an important role in immune activity, mediating the release of cytokines and the expression of receptors engaged in the immune response [87]. The epithelial layer also includes the goblet cells secreting mucus, Paneth cells synthesising defensins, enterochromatophilic cells releasing hormones and neuropeptides, and M cells that capture antigens from the intestinal lumen. The epithelial integrity and as well as selective permeability are dependent on appropriate connections between cells, mainly tight junctions, including claudins, occludins, protein junctional adhesion molecules, and tricellulins [88].

Oral iron supplementation contributes to lipid peroxidation of cellular membranes and disrupts energy processes in the cell due to mitochondrial damage, as well as promoting endoplasmic reticulum dysfunction [89]. Consequently, these cellular disorders trigger death pathways, thereby destroying the mechanical intestinal barrier and increasing permeability, known as a leaky gut [47]. There are also changes in the enterocyte shedding-proliferation axis [90]. This impairment of the intestinal barrier increases the risk of exposure to bacterial endotoxins (e.g., LPS), leading to metabolic endotoxemia and microinflammation (gut-derived inflammation). LPS binding to its receptor complex on macrophages results in markedly increased production of pro-inflammatory cytokines such as IL-1β, IL-6, IL-12, IFN-β or IFN-γ, which may worsen intestinal inflammation in IBD patients. This impaired integrity has been demonstrated in an in vitro study with Caco-2 cells exposed to iron [91].

The proper functioning of the intestinal barrier also involves short-chain fatty acids (SCFAs) including butyrate, acetate and propionate, produced by the microbiota in the colon through the anaerobic fermentation of indigestible polysaccharides such as dietary fibre and resistant starch. There are two main pathways for the conversion of butyrate–CoA into butyrate. The first pathway includes a two-step reaction using butyrate kinase and phosphate butyryltransferase. The second pathway is a single-step reaction conducted by butyryl–CoA: acetate Co-A transferase [92]. SCFAs are the main source of energy to colonocytes and modulate the immune response through inhibition of the LPS-induced NF-κB pathway, and reduced production of pro-inflammatory cytokines and chemokine by the epithelial intestinal cells (EIC). Besides, SCFAs increase the secretion of antimicrobial peptides (LL-37 and CAP-18) and IL-18, a cytokine that maintains homeostasis in EIC, thereby protecting against colitis [93,94].

The influence of oral iron supply on SCFAs production remains still unclear. Some research indicates that iron can contribute to increases in gut SCFAs production, thus positively affecting gut health [95]. Some in vitro studies compared the impact on SCFAs production between the normal iron condition and iron deficiency, which does not fully reflect increased iron content in the gut lumen on SCFAs production [96]. Another study investigated the influence of oral iron supplementation on colitis exacerbation and the composition of the gut microbiome. A decreased abundance of SCFA-producing genera was shown without assessing the levels of faecal SCFAs [97]. However, the literature also provides evidence for the adverse impact of oral heme iron on murine colitis model, reducing the level of butyrate production and expression of butyrate kinase, phosphate butyryltransferase and the α subunit of butyryl–CoA: acetate Co-A transferase [92].

In conclusion, the loss of intestinal barrier integrity is an early event which contributes to chronic inflammation (Figure 3).

### 3.5. Impact of Iron on the Microbiota in IBD Patients

Besides the unfavourable impact of iron on the intestinal epithelial barrier via oxidative stress, iron may also affect the gut microbiota [98]. As mentioned above, the altered gut microbiota is a crucial factor in driving inflammation in IBD but it is unclear if dysbiosis is the cause or the outcome of mucosal inflammation. Numerous studies have indicated differences in the composition and diversity of the intestinal microbiota among IBD patients in comparison to healthy individuals. Characteristic changes observed in patients with IBD include increased bacteria such as Proteobacteria, Fusobacterium species, and *Ruminococcus gnavus* and in turn, decreased protective groups such as Lachnospiraceae, Bifidobacterium species, *Roseburia*, and *Sutterella* [99,100]. Iron leads to a shift in the microbiota composition and exacerbation of dysbiosis in IBD. Low iron bioavailability results in a high concentration in the intestinal lumen and accessibility to the gut microflora. Lee et al. [101] designed an open-labelled clinical trial with iron-deficient participants with or without IBD and performed pre-iron therapy and post-iron therapy measurements whereby the individuals served as the controls. This study compared the effects of *per oral* (p.o) versus *intravenous* (i.v) iron replacement therapy (IRT), indicating that oral iron supplementation decreased the diversity of intestinal microflora in patients with IBD and ID, especially *Faecalibacterium prausnitzii*, *Ruminococcus bromii*, *Dorea* sp. and *Collinsella aerofaciens*. It is worth noting that dysbiosis aggravated by iron can also be a causative factor disrupting the integrity of the EIC and promoting an oxidative pro-inflammatory microenvironment.

In summary, the need for iron supplementation in IBD patients who additionally suffer from anaemia exacerbates the dysbiosis caused by the disease, thus worsening their clinical condition.

## 4. Colorectal Cancer

Colorectal cancer (CRC) is the third most commonly diagnosed cancer globally and the second major cause of mortality (935,000 deaths in 2020) [76]. It affects mainly older adults from highly developed countries, as it is a multifactorial disease closely associated with lifestyle and risk factors like obesity, lack of physical activity, alcohol consumption, cigarette smoking or low fibre, fruit and vegetable supply [102]. The American Cancer Society estimates the lifetime risk of developing colorectal cancer is about 4.3% for men and 4% for women. Increasing research indicates that iron supplementation and the resulting changes in the composition of the intestinal microbiota are essential factors causing CRC development.

### 4.1. Impact of Heme Iron on Cancerogenesis

This section discusses the influence of heme iron on CRC development [103]. The relationship between CRC and non-heme iron has not been established and requires further investigation. The International Agency for Research on Cancer classified the consumption of processed meat as carcinogenic and red meat as probably carcinogenic in humans in 2015. There are several reasons for such classification, but most importantly, heme iron plays a crucial role in promoting colorectal carcinogenesis. Heme iron contributes to DNA damage through the increased formation of lipid peroxyl radicals (MDA and 4-hydroxynonenal) and catalysing ROS production, as well as increasing the endogenous formation of *N*-nitroso compounds (NOCs), all of which are cytotoxic and genotoxic. There is a growing body of evidence that heme iron indirectly contributes to a weakened gut mucosal barrier and proliferation of the colon epithelium. This process is stimulated by a large amount of heme iron found in red meat, which is an essential factor in the development of pathogenic microbiota, reducing the number of probiotic bacteria such as lactobacilli that do not require iron. These pathobionts are sulphate-reducing and mucin-degrading bacteria, especially *Akkermansia mucinphila*, which reduces disulphide bonds between mucin proteins, thus reducing the mucus layer. This process increases the intestinal wall permeability to luminal cytotoxic compounds and products of bacterial degradation, which in turn, leads to compensatory hyperproliferation, hyperplasia and finally, neoplasm [104,105].

Bastide et al. showed that heme iron plays a leading role in mucin-depleted foci (MDF) formation [106]. Moreover, some gut strains, such as *Bacteroides fragilis*, can use heme iron directly [107]. Various studies have shown the effect of iron intake on intestinal microbiota composition; for example, the six months consumption of iron-supplemented biscuits by anaemic African children caused a potentially more pathogenic gut microbiota profile and increased inflammation. There was also a significant growth of enterobacteria and a reduced amount of lactobacilli when compared to the control group receiving non-supplemented biscuits [46]. In Kenya, infants were given iron-fortified micronutrient powder, which also caused significant changes in the intestinal microbiota, increasing enterobacteria and decreasing lactobacilli, resulting in inflammation as evidenced by an increase in faecal calprotectin [69]. The described mechanisms are summarised in Figure 4.

### 4.2. Dysbiosis and Cancerogenesis

The bacterial strains that dominate in the colon during dysbiosis contribute to DNA damage, suppression of apoptosis by affecting multiple signalling pathways, and the production of pro-inflammatory cytokines or toxic metabolites. The microbes activate the NF-kB (nuclear factor-κB) pathway via the host Toll-like receptors (TLRs), initiating carcinogenesis due to the inhibition of apoptosis in enterocytes. Moreover, the activated NF-kB pathway causes an increase in the production of pro-inflammatory cytokines, TNF-α, IL-1, IL-6, IL-8 and anti-apoptotic genes [108]. NF-kB activation is also possible via the increased production of secondary bile acids, which cause an increase in reactive oxygen, the direct activator of the NF-kB signalling pathway [109].

The adenomatous polyposis coli (Apc) gene is mutated in both types of CRC, familial sporadic and colitis-associated CRC. Apc is the primary regulator of the Wnt/β-catenin signalling pathway responsible for the physiological maturation of enterocytes. The activation of Wnt signalling leads to the nuclear accumulation of β-catenin and begins continuous stimulation of the transcription T-cell factor/lymphoid enhancer factor (TCF/LEF), in turn, activating the target genes, c-myc and cyclin D1, initiating uncontrolled cell proliferation. Iron and iron-related-dysbiosis can increase Wnt signalling following the loss of Apc function [99,100].

The activation of the Wnt/β-catenin pathway is also possible through substances produced directly by the bacteria that colonise the gut. For example, studies have shown that *Bacteroides fragilis* toxin enhances cell signalling via the Wnt/β-catenin pathway [97]. Furthermore, in a mouse model, *Fusobacterium nucleatum*, a gram negative oral anaerobe, activates the E-cadherin/β-catenin pathway via the adhesin factor FadA leading to upregulation of chk2, increasing DNA damage and enhancing cancer tumor growth [101]. The described mechanisms are summarised in Figure 5.

### 4.3. Microbiota Composition and Bacterial Metabolites

Bacterial drivers of colorectal cancer are defined as intestinal bacteria with carcinogenic characteristics that may initiate tumour development. The accumulation of specific genera of the bacteria has been observed in colon cancer and currently there are three strains of bacteria with proven cancerogenic potential: enterotoxigenic *Bacteroides fragilis* (ETBF), Fusobacterium and certain *Escherichia coli* strains [110]. ETBF produces *Bacteroides fragilis* toxin (BFT) also known as fragilysin, which upregulates spermine oxidase (SMO) in colonic epithelial cells, causing an increase in ROS. ETDF also stimulates the production of IL-17 by Th-17 cells to support the growth and survival of cancer cells and suppresses T-cell proliferation [111]. Fusobacterium also increases IL-17, TNF-α, IL-6, IL-8, and IL-12 production and is associated with high cyclooxygenase 2 (COX-2) activity and a 3.5-fold increased risk of colonic adenomas [112].

The most genotoxic substance involved in CRC development, colibactin, is produced by the pathogenic *E. coli* strains possessing the polyketide synthase (pks) island. Colibactin contributes to chromosome aberrations or double-strand DNA breaks (Dubinsky b.d.). It also suppresses the mutL homologue 1 (MLH1), the mismatch repair protein. There are suggestions that the use of inhibitors of colibactin synthesis could stop the proliferation of tumour cells and be used as a new form of treatment [113].

The *E. coli* strains in CRC are more invasive than strains in other diseases. They support SUMO (small ubiquitin-like modifier) conjugation to the p53 tumour suppressor protein and cause cell senescence [109]. *Enterococcus faecalis* can also produce secondary bile salts and hydrogen sulphide, promoting inflammation and cancerogenesis. Hydrogen sulphide is toxic to the intestinal epithelium and causes damage to it, as well as contributes to the formation of mutations [114].

### 4.4. Comparison of the Intestinal Microflora of Healthy Subjects and CRC Patients

Studies in mice have proved the differences between the healthy versus CRC gut microflora. When compared to a healthy gut, a CRC gut is more abundant in *E. coli* and Proteobacteria with fewer Bacteroidetes [111]. Another study has revealed a profusion of Fusobacterium, Campylobacter and Leptotrichia in CRC patients. It is worth noting that these are oral bacteria; therefore, the critical question is whether these bacteria are from the oral cavity, belong to the colon microbiota, or are a cancer strain [115].

### 4.5. Driver–Passenger Model

Tlajsma et al. developed a model in which pathogenic bacteria (termed bacterial drivers) cause permanent changes in the intestinal epithelium leading to CRC development. The alteration of the environment enables colonisation by the colon passenger bacteria, which take advantage of the tumour microenvironment. By competing with driver bacteria, they reduce their abundance, suggesting that the microenvironment changes as the tumour progresses. CRC development increases the number of passenger bacteria while the number of driver bacteria decreases [110]. By proving that passenger bacterial growth depends on CRC metabolites, Gorza et al. confirmed the driver–passenger hypothesis [116]. Research by Wang et al. made it possible to distinguish species referred to as drivers and passengers, with potential driver bacteria including Bacillus, Bradyrhizobium, Methylobacterium, Streptomyces, Intrasporangiaceae and Sinobacterace located on off-tumour sites. The on-tumour sites harboured fourteen species of potential passengers, Fusobacterium, Campylobacter, Streptococcus, Schwartzia, Parvimonas, Dethiosulfatibacter, Selenomonas, Peptostreptococus, Leptotrichia, Granulicatella, Shewanella, Mogibacterium, Eikenella, and Anaerococus. The authors suggest that the driver or passenger species can be used as a biomarker for estimating the risk of initiation of CRC or patients with CRC [117].

### 4.6. Probiotic Bacteria and Butyrate

Probiotic bacteria maintain favourable intestinal microbiota by preserving the integrity of the intestinal barrier and reducing bacterial translocation. Attempts have been made to use probiotics in the treatment of colorectal cancer and to reduce the risk of its development. The analysis of randomised trials shows that using probiotics, especially *Lactobacilli* and *Bifidobacterium* species, reduces postoperative complications and accelerates patient recovery [118]. Faghfoori et al. proved that some *Bifidobacteria* species produce substances that affect the activation of apoptotic pathways in human colorectal cancer cells (HT-29 and Caco-2 cell lines) but highlighted the need for further research in this area [119].

Low fibre intake is one of the risk factors for CRC development. During fibre fermentation, the gut microbiota produces SCFAs, such as butyrate, the crucial energy source for intestinal epithelial cells. Besides, butyrate over-activates the Wnt signalling pathway and promotes cancer cell apoptosis. It has been shown that CRC patients have fewer butyrate-producing bacteria than healthy controls [120]. Research in mice has demonstrated that butyrate inhibits the proliferation of cancer cells without affecting healthy colon cells. Moreover, butyrate reduces the number of free radicals [117]. The knowledge in this area can contribute to proposing the appropriate composition of new probiotic therapy that can be used in treating CRC [111].

### 4.7. Iron Deficiency (ID) and Supplementation

Initially, in the course of colorectal cancer, excess iron contributes to its progression. ID occurs as the disease progresses due to frequent bleeding from the gastrointestinal tract, reduces hematopoiesis and decreases the immune response, allowing tumour cells to survive. Furthermore, ID contributes to the modification of macrophage polarisation and Treg populations, favouring a carcinogenic tumour immune microenvironment. These processes often contribute to a poor treatment response and decreased survival after surgery [103]. Studies by Kam et al. proved that intravenous iron administration before surgery increases the level of Hb, lowering the incidence of blood transfusion and reducing the risk of complications associated with the surgery [121]. Another study has shown that postoperative intravenous iron administration to anaemic patients increases the concentration of Hb without postoperative complications [122]. Oral iron supplementation is not recommended due to contributing to the progression of CRC, as it increases the iron concentration in the colon and causes significant gastrointestinal side effects such as diarrhoea, constipation, nausea or abdominal pain [9].

## 5. Obesity

Obesity, called an epidemic of the 21st century, is a condition of pathological adipose tissue accumulation in the body and is associated with many disorders such as cardiovascular disease, Type 2 diabetes, chronic kidney disease, retinopathy and several cancers [123]. Due to the chronic inflammation accompanying obesity, it can be considered a disease by itself. The devastation of physiological processes caused by obesity also affects iron management, with numerous studies confirming that obese individuals are characterised by a lower iron concentration when compared to people of a normal body weight [123,124,125].

### 5.1. Adipose Tissue as an Endocrine Organ

Apart from providing thermal isolation and storing energy, adipose tissue is the largest endocrine organ in terms of mass. Its cell composition, tissue weaving, and ability to produce and secrete adipokines differ depending on its location in the body [126]. Visceral adipose tissue is more metabolically active, better vascularised and has more macrophages than subcutaneous fat [126,127]. Adipocytes secret pro-inflammatory adipokines and cytokines, such as adiponectin, leptin, resistin, PAI-1, TNF-α, IL-6, CRP, and more cytokines involved in insulin resistance [127].

### 5.2. Chronic Low-Grade Inflammation: An Inherent Consequence of Obesity

Currently, obesity is considered a low-grade systemic, chronic inflammation. The fact that adipose tissue is significantly involved in the production of pro-inflammatory cytokines makes it an indispensable marker of obesity [126].

A high-calorie diet, rich in fat and sugar, causes dysbiotic changes in the intestinal microbiota, becoming much less biodiverse with an increased ratio of Gram-positive Firmicutes to Gram-negative Bacteroidetes, thereby increasing energy production from indigestible carbohydrates [128,129]. Moreover, dysbiosis contributes to an increased “porosity” of the intestinal wall, which leads to expanded contact of microorganisms with the intestinal mucosa, activation of TLRs and inflammation. Increased intestinal permeability also promotes an increase in blood lipopolysaccharides (LPS), which enhance the secretion of IL-6 and CRP [129]. Dysbiosis in obese people also enhances further weight gain. The relationship between microbiota and obesity is two-sided. In studies in mice, the transfer of microbiota from obese to lean animals resulted in an increase in energy absorption from food and weight gain. Conversely, the use of prebiotics in mice to improve microflora decreased gut permeability and overall inflammation. Moreover, as a result of prolonged antibiotic therapy in the treatment of endocarditis, patients gained weight. Bell et al. emphasised the role of dysbiosis in the development of obesity, suggesting that only the combination of genes related to obesity with dysbiosis leads to fat deposition and inflammation [128]. The high ratio of saturated to unsaturated fatty acids in the diet promotes TLR activation and thus the expression of IL-6, TNF-α, and chemokines. Another issue is adipocyte hyperplasia as the need for fat storage increases. Since the possibilities of their growth are limited, the cells may break, promoting inflammation. Visceral adipocytes are even more unstable because they are exposed to sudden changes in pressure when coughing or exercising [127]. Another mechanism observed in obese patients is the infiltration of macrophages into adipose tissue, especially visceral fat. They are formed because of the transformation of monocytes from the bloodstream and their concentration in adipose tissue positively correlates with the amount of fat. Macrophages are believed to be the main source of inflammatory cytokines such as TNF-α, produced by adipose tissue [123,126].

### 5.3. Iron Deficiency (ID) Is Common in Obese Individuals

ID often accompanies obesity, with many studies showing a negative correlation between body mass index (BMI) and serum iron levels [124,130,131]. One of the mechanisms leading to ID in obesity is the increase in the expression of hepcidin, a protein involved in regulating iron homeostasis [123]. In a Swiss study, overweight children consumed similar amounts of bioavailable iron in their diet as children with a normal BMI but had lower iron levels and higher hepcidin, Il-6, CRP, and leptin concentrations [132]. Although the liver is the main site of hepcidin production, it is also expressed in adipose tissue, the weight of which in obese people can be twenty times greater than that of the liver [133]. Many factors stimulate its expression but apart from iron overload, other mechanisms are also of particular importance in obesity. These include cytokines secreted by adipose tissue, such as IL-6, IL-1, TNF-α, CRP, and leptin [123]. Additionally, a high leptin level stimulates the transcription of hepcidin genes via the JAK/STAT signalling pathway in hepatocytes. In addition, a study in mice showed that a diet that provides the body with excess nutrients could induce endoplasmic reticulum stress, which also increases the secretion of hepcidin [134]. Moreover, it often leads to damage of hepatocytes and further overproduction of hepcidin by the Stat3 and C/EBP pathways [124]. Subsequently, hepcidin is degraded by binding to ferroportin, thereby preventing the transport of iron into the bloodstream, thus reducing its concentration in the body. As a result, iron provided in the diet, despite its bioavailable form, is not absorbed [123,124,125].

Ferroportin degradation also disrupts iron transport from the macrophages of the spleen, liver and bone marrow into the bloodstream, resulting in an accumulation of iron in the organs mentioned above [135]. Furthermore, hemojuvelin expression occurs in adipose tissue, which stimulates the local expression of hepcidin via the BMP–HJV pathway. Its concentration increases significantly in obese patients affecting general iron homeostasis [134]. Numerous studies show that disturbances in iron balance in obesity also occur independently of hepcidin. The impairment of iron uptake by enterocytes is caused by the disturbed expression of oxidoreductases, despite the deficiency of iron in enterocytes [136,137]. A study in mice showed that a diet rich in fat decreased mRNA levels of duodenal cytochrome B (DcytB) oxidoreductase and hepcidin, leading to the impairment of redox processes necessary for the transport of iron through enterocyte membranes [136]. The above-described mechanism involved in the development of iron deficiency in people with obesity is pictured in Figure 6.

### 5.4. Dysmetabolic Iron Overload Syndrome (DIOS)

The disturbances in iron metabolism in one-third of obese patients take the form of hyperferritinemia. Typical symptoms are increased ferritin levels and normal or slightly increased transferrin saturation. The condition is usually associated with metabolic syndrome (MS) components: high blood pressure, high blood triglycerides, low levels of HDL cholesterol and insulin resistance. The increase in ferritin positively correlates with the number of MS characteristics. Both ID and DIOS are characterised by an increased concentration of hepcidin and a lower expression of ferroportin, suggesting that both disease entities are different symptoms of the same underlying pathology. Factors such as age, gender, and the degree of BMI increase the influence of the form of iron disturbance in obesity. ID occurs in adolescents and adults with morbid obesity, in which iron is lost as a result of menstruation or inflammation of adipose tissue, whereas iron accumulation becomes a problem in postmenopausal women and insulin-resistant men. Contrary to many diseases in the case of obesity, the literature provides little data on iron supplementation, its effectiveness and its impact on the microbiota. This issue seems to be significant considering the scale of the problem [123,138].

### 5.5. Treatment of Iron Deficiency (ID) in Obesity

There is little well-designed research regarding oral iron supplementation in obese subjects. Nevertheless, Hurrel et al. [135] speculate that it is problematic due to impaired iron absorption caused by general inflammation and rising hepcidin levels. However, Rabindrakumar et al. [139] reported an increase in iron concentration in the intestines accompanying daily supplementation which may stimulate hepcidin expression. Some studies suggest that weight loss reduces inflammation and hepcidin levels and improves iron absorption from the diet. There was also an increase in transferrin saturation and the restoration of normal iron homeostasis [123,135,140]. Due to the proven relationship between dysbiosis, obesity, and iron status, one of the methods of treating ID in obese patients has become probiotic therapy. Several studies have been conducted with promising results. Iron management has been proven to improve with single strain therapy through various mechanisms. A 2020 study described the function of probiotics as iron carriers, demonstrating that probiotics convert inaccessible forms of iron into absorbable forms, producing metabolites that stimulate iron absorption and reduce the Fe accumulation in the liver [141]. A multi-strain supplementation study showed reductions in hair iron and blood FAM levels leading to the conclusion that Fe shifted from hair to bone marrow. However, too few volunteers took part in the study to confirm the effectiveness of the therapy in this case [125]. The described results show a relationship between the microbiota status in obese people and iron metabolism.

## 6. Summary

The side effects of iron supplementation are indisputable and often irreversible, whether iron is delivered to the gastrointestinal tract or the blood (Figure 7).

In the blood, only protein-bound iron is safe. If the transferrin and ferritin binding capacity is exceeded, the serum’s free iron concentration, the labile Fe pool increases, generating reactive oxygen species (ROS) [31,33]. Subsequently, lipid and amino acid peroxidation, enzyme denaturation, polysaccharide depolymerisation and DNA damage lead to endothelial cell dysfunction [10,18,39,40,41,42,43]. Reactive oxygen species are also critical for activating the physiological signalling pathway NF-κB and producing pro-inflammatory cytokines such as IFN-γ, TNF-α, and iNOS. All the processes mentioned lead to systemic inflammation.

Oral iron supplementation activates ROS production simultaneously in the gut lumen and the enterocytes. It is the beginning of the local inflammatory process leading to damage to the intestinal wall integrity and resulting in a leaky gut syndrome. The increased permeability of the intestinal wall leads to the leakage of metabolites and bacterial toxins into the blood resulting in endotoxaemia and contributing to systemic inflammation. In the case of IBD, it worsens clinical symptoms [47,88].

Iron in the intestinal lumen directly activates the NF-κB pathway and stimulates the expression of pro-inflammatory cytokines and iNOS. This contributes to the deterioration of symptoms and is the reason why oral iron administration in inflammatory bowel disease (IBD) is not recommended [82,83].

The excess unabsorbed iron supplemented orally passes through the colon and becomes an essential growth factor for nearly all pathogenic bacteria, fungi and protozoa, as well as for all neoplastic cells [32]. It reduces the abundance of beneficial microbes simultaneously with an increased abundance of deleterious microbes, leading to dysbiosis [66,68].

The numerical advantage of unfavourable bacterial strains leads to increased energy absorption from the food and increased body weight, which is particularly detrimental for obese individuals [128].

## 7. Conclusions

Despite severe side effects, supplementation of iron deficiencies in the case of anaemia is necessary. Well-planned research, specific to each patient category and disease, is needed to find measures and methods to optimise iron treatment and reduce adverse effects.

## Figures and Tables

**Figure 1 nutrients-14-03478-f001:**
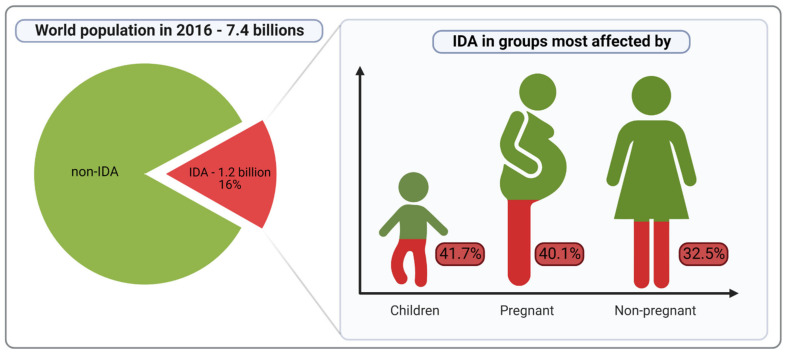
Occurrence of iron deficiency anemia. IDA—iron deficiency anemia; Non-IDA—anemia without iron-deficiency.

**Figure 2 nutrients-14-03478-f002:**
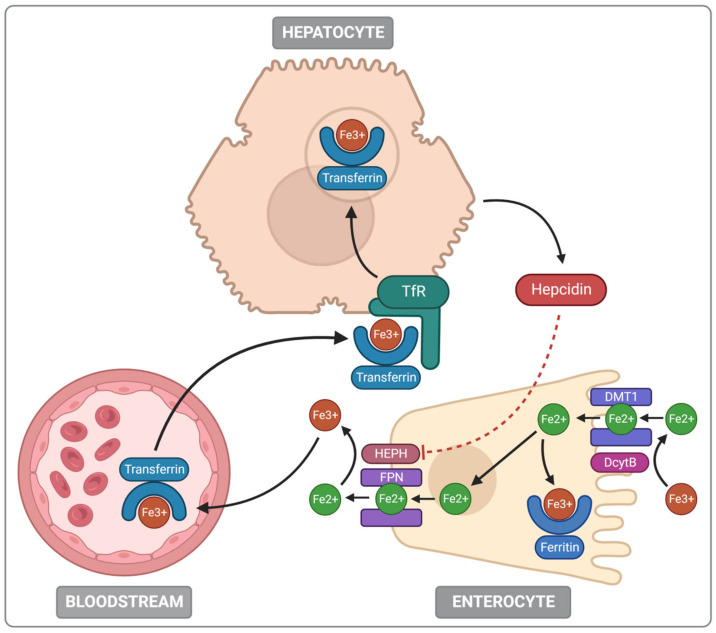
Iron metabolism. DcytB—duodenal cytochrome B; DMT1—divalent metal-ion transporter-1; FPN—ferroportin; HEPH—hephaestin; TfR—transferrin receptor; dashed line–inhibition.

**Figure 3 nutrients-14-03478-f003:**
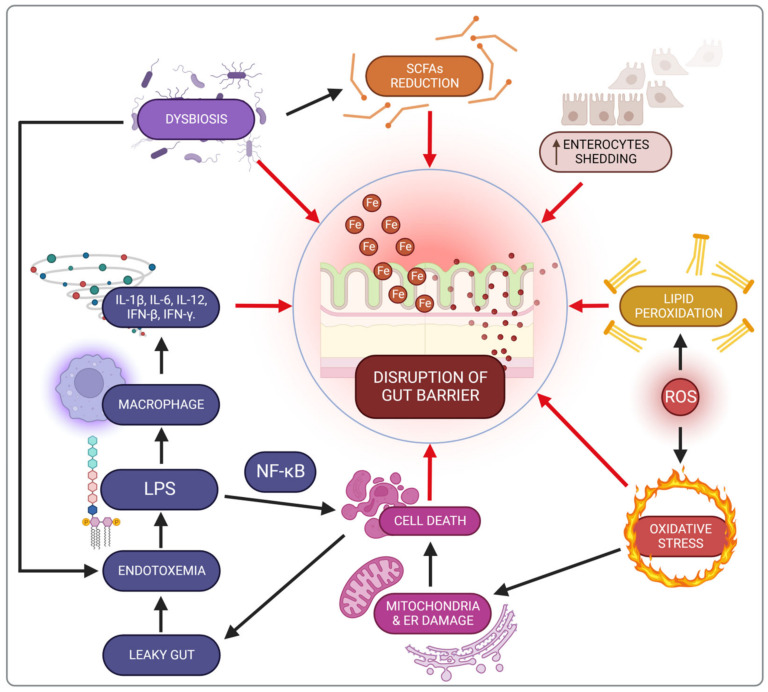
Iron-related gut barrier disruption mechanisms. SCFAs—short-chain fatty acids; IFN-β—interferon β; IFN-γ—interferon-γ; IL-1β—interleukin 1β; IL-6—interleukin 6; IL-12—interleukin 12; LPS—lipopolysaccharide; NF-kβ—nuclear factor kappa-light-chain-enhancer of activated B cells; ROS—reactive oxygen species.

**Figure 4 nutrients-14-03478-f004:**
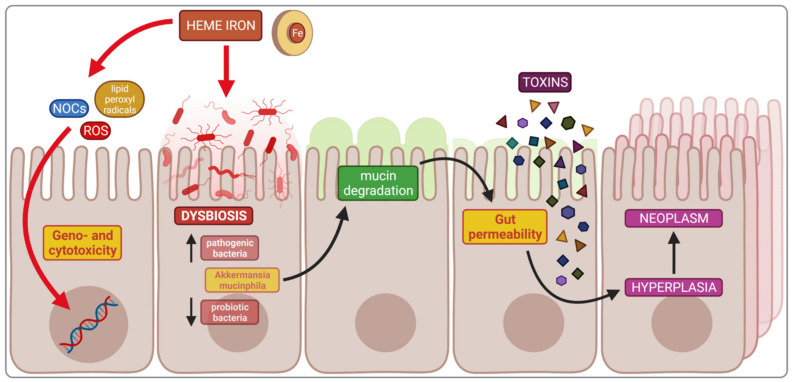
Mechanisms of heme iron induction of colorectal carcinogenesis. NOCs—*N*-nitroso compounds; ROS—reactive oxygen species.

**Figure 5 nutrients-14-03478-f005:**
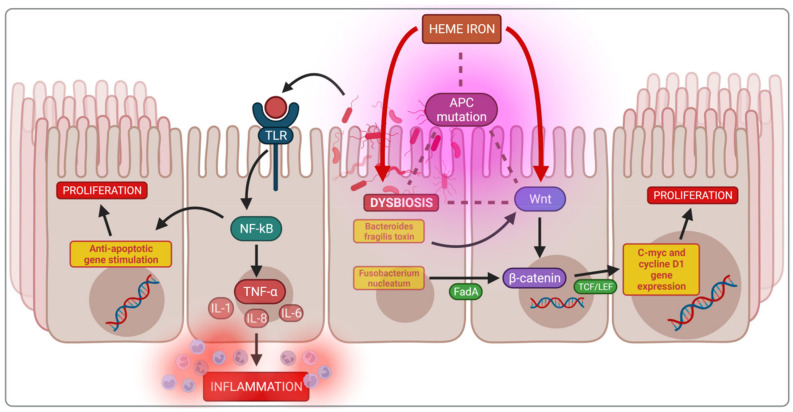
Mechanisms of heme iron induction of colorectal carcinogenesis. APC—adenomatous polyposis coli gene; C-myc, c-Myc–oncogenes; FadA—Fusobacterium nucleatum adhesin; IL-1—interleukin 1; IL-6—interleukin 6; IL-8—interleukin 8; NF-kβ—nuclear factor kappa-light-chain-enhancer of activated B cells; TCF/LEF—T-cell factor/lymphoid enhancer factor family; TLR—Toll-like receptor; TNF-α—tumor necrosis factor α; Wnt—Wnt signaling pathways.

**Figure 6 nutrients-14-03478-f006:**
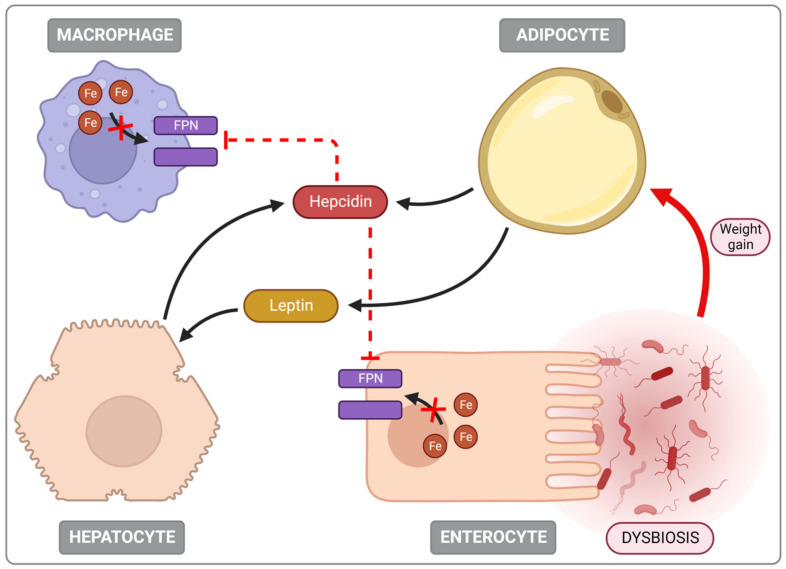
Iron deficiency in people with obesity. FPN—ferroportin, dashed line—inhibition.

**Figure 7 nutrients-14-03478-f007:**
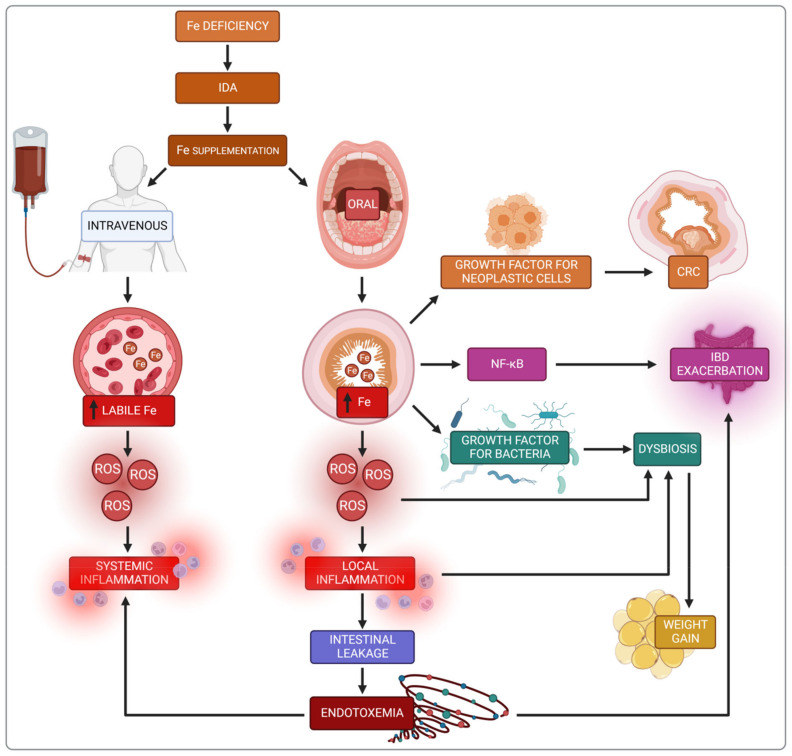
Impact of oral and intravenous iron supplementation on development and course in chosen clinical conditions. IDA—iron deficiency anemia; CRC—colorectal cancer; IBD—inflammatory bowel disease; NF-kβ—nuclear factor kappa-light-chain-enhancer of activated B cells; ROS reactive oxygen species.

**Table 1 nutrients-14-03478-t001:** Impact of oral iron supplementation on gut microbiota composition (↓—decrease, ↑—increase).

Bacteria	Iron Supplementation
Phylum: Firmicutes	↓
Genus: Enterococcus	↑
Genus: Lactobacillus	↓
Genus: Roseburia	↑
Genus: Clostridium	↑
Phylum: Proteobacteria	↑
Family: Enterobacteriaceae	↑
Species: E. coli	↑
Genus: Salmonella	↑
Genus: Shigella	↑
Genus: Citrobacter	↑
Order: Bacteroidales	↑
Genus: Bacteroides	↑
Genus: Campylobacter	↑
Genus: Bifidobacterium	↓
Genus: Prevotella Genus: Rothia	↓ ↓

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
