# Peer review of "The Dark Side of Iron: The Relationship between Iron, Inflammation and Gut Microbiota in Selected Diseases Associated with Iron Deficiency Anaemia—A Narrative Review"

_nutrients, 2022, doi:10.3390/nu14173478_

Round 1

Reviewer 1 Report

Thank you for the opportunity to review this interesting article, which is a narrative review of iron deficiency anaemia and treatments in common inflammatory linked conditions. The article is a thorough and very detailed review of mechanisms and current literature. It is quite long but it is clearly written and nicely illustrated.  I have just a few minor suggestions:

Abstract: i think the search strategy needs to come before the conclusion regarding the need for 'well planned research, specific to ...' in the abstract. This search strategy is your method and so would not be at the end. 

I can't see that your search strategy is mentioned in the main body of the article. I am assuming that everything from section 1.1 onwards is related to information you found in the literature search. I think you need to describe your search strategy, as you have in the abstract, at least in the introduction or it could have it's own short section, before section 1.1 

Page 2: you mention that a holistic approach to management of ID is available - could you describe what this entails please for example, dietary intake. 

Page 7: need to add the reference for the prevalence of IDA in IBD please

Please  re-check the use of the word 'pule' in the article. Do you mean pool as is used towards the end of the article?

Thank you, i look forward to reading a revised version of this article. 

Reviewer 2 Report

Summary

This review article discusses the effects of iron consumption on inflammation and microbiota, focusing on diseases related to iron deficiency. The authors have subdivided the sections focusing on 1.) The physiological role of iron in the body and an introduction to the microbiota 2.) Inflammatory Bowel Diseases 3.) Cancer and 4.) Obesity. The authors utilized multiple databases, culminating in an extensive review using 130 articles. This is a needed review article that nicely summarizes some of the recent advances in knowledge and highlight the areas that need to be further investigated.

Minor Comments

1.     Minor grammatical errors such as a missing word on line 76. I believe it should say “ID and excess iron lead to…”. Write out World Health Organization line 41 in the introduction. Add a space after the period on line 119. Missing period on line 133 after [32].

2.     Add ; before the word therefore followed by a , after.

3.     Section 1.2, line 118 add “which are responsible for post-transcriptional…”

4.     Section 1.2, line 102-103 have the definition of ferritin directly after the word

5.     Section 1.4, line 151-152. Perhaps could be changed to say “increased peroxidation of lipids leading to the destruction of cell membranes”

6.     Section 1.5, line 176. Lactobacillus and Bifidobacterium should be in italics

7.     Section 2.5, line 294 missing a comma after adhesion molecules

8.     Section 3.3, line 396-397. Interleukins are shortened to IL as in IL-6, IL-8

9.     Section 4.2, line 531 should be IL-6 and CRP should be written out.

10.  Numerous cases of a missing comma when listing multiple items

Major Comments

1.     Section 1.5, line 190-191 is a bit out of place. Are the authors trying to say that the African children that consumed iron-fortified biscuits presented with increased calprotectin in their stool?

2.     Fix the alignment of the columns in Table 1..

3.     The section on SCFA in IBD (lines 307-317 in section 2.5). The authors suggest that oral iron consumption reduces SCFA available in the gut but do not explain how. Is this an indirect or direct effect of iron consumption on the microbiota? Perhaps this section should be with the microbiota in section 2.6.

4.     Consistency with regards to the sections needs to be fixed. Only section 4 on Obesity has a bolded Heading. Either all of the sections should have a short intro with a bolded heading or not.

Reviewer 3 Report

This manuscript reviewed the potential side effects of iron supplementation since iron supplementation alters inflammatory responses and microbiomes. In overall, it is a well-summarized review, but some points may be updated.

1. Iron supplementation also alters the absorption, distribution, and metabolism of other divalent metals which may affect systemic inflammation and microbiomes. Therefore, authors should acknowledge iron supplementation and other metals (i.e. Fe X Cu - PMID; 33801587, 29546308, 28619730, 27537180).

2. Please avoid using pronouns (i.e. L 15 Unfortunately, this...).

3. IRE paragraph is extremely short. Please give more information (L117-120).

4. [L174] Proper superscript is required for Fe3+.

5. [L348] A reference is required for death counts.
